# Porous Fly Ash/Aluminosilicate Microspheres-Based Composites Containing Lightweight Granules Using Liquid Glass as Binder

**DOI:** 10.3390/polym14173461

**Published:** 2022-08-24

**Authors:** Olga Miryuk, Roman Fediuk, Mugahed Amran

**Affiliations:** 1Department of Construction and Building Materials Science, Rudny Industrial Institute, Rudny 111500, Kazakhstan; 2Polytechnical Institute, Far Eastern Federal University, Vladivostok 690922, Russia; 3Peter the Great St. Petersburg Polytechnic University, St. Petersburg 195251, Russia; 4Department of Civil Engineering, College of Engineering, Prince Sattam Bin Abdulaziz University, Alkharj 16273, Saudi Arabia; 5Department of Civil Engineering, Faculty of Engineering and IT, Amran University, Amran 9677, Yemen

**Keywords:** liquid glass, thermal energy waste, lightweight concrete, porous filler, heat-insulating materials

## Abstract

The modern energy-saving vector of development in building materials science is being implemented in a complex way through the development of new heat-insulating materials with the simultaneous exclusion of low-ecological cement from them. This article presents the results of the development of resource-saving technology for a heat-insulating composite material. The research is devoted to the development of scientific ideas about the technology and properties of effective cementless lightweight concretes. The aim of the work is to create a heat-insulating composite material based on porous granules and a matrix from mixtures of liquid glass and thermal energy waste. The novelty of the work lies in establishing the patterns of formation of a stable structure of a porous material during thermal curing of liquid glass with technogenic fillers. Studies of liquid glass mixtures with different contents of fly ash and aluminosilicate microspheres revealed the possibility of controlling the properties of molding masses in a wide range. To obtain a granular material, liquid glass mixtures of plastic consistency with a predominance of aluminosilicate microspheres are proposed. The matrix of composite materials is formed by a mobile mixture of liquid glass and a combined filler, in which fly ash predominates. The parameters of heat treatment of granular and composite materials are established to ensure the formation of a strong porous waterproof structure. The possibility of regulating the structure of composite materials due to different degrees of filling the liquid glass matrix with porous granules is shown. A heat-insulating concrete based on porous aggregate has been developed, characterized by the genetic commonality of the matrix and the granular component, density of 380–650 kg/m^3^, thermal conductivity of 0.095–0.100 W/(m °C) and strength of 3.5–9.0 MPa, resistance under conditions of variable values of humidity and temperature. A basic technological scheme for the joint production of granular and composite materials from liquid glass mixtures is proposed.

## 1. Introduction

Modern construction requires durable materials that combine high heat-shielding properties with structural stability [1]. High-energy efficiency of construction can be achieved by the wide use of lightweight concrete [2]. The advantages of lightweight concrete are to reduce the mass of structures and the consumption of steel reinforcement in reinforced concrete foundations, improve thermal performance and reduce the cost of construction objects [3,4,5,6]. Reducing the mass of structural elements of buildings makes it possible to implement complex architectural and space-planning solutions [7]. The possibility of using lightweight concrete for additive construction technologies is shown [8]. A long experience in the operation of buildings and structures confirms the durability, environmental safety and operational reliability of lightweight concrete [9]. Expanded clay concrete is a common type of lightweight concrete [10,11,12]. Increasing requirements for the heat-shielding properties of building envelopes have reduced the consumption of expanded clay concrete [13]. Another reason for the decrease in the share of expanded clay concrete in construction is the high cost of expanded clay compared to dense aggregate, the increased consumption of cement in structural lightweight concrete [14].

The reserves for increasing the efficiency of lightweight concretes lie in the use of cheaper porous aggregates, reducing the thickness of the coating of aggregate grains with cement paste due to effective binders [15]. The characteristics of lightweight concrete depend on the quality of the porous aggregates [16]. For the development of lightweight concrete technology, research aimed at reducing the density of aggregates and reducing the energy intensity of production is relevant [4,17,18,19,20,21]. There is an extensive raw material base for the production of porous aggregates [22]. The resource saving technology of porous aggregates is ensured by the use of technogenic materials of mineral and organic origin [23,24,25,26,27]. A lot of positive experience has been accumulated in obtaining lightweight concrete using ash from coal combustion [28]. Ashes formed at thermal power plants are used as a fine aggregate, as part of raw mixtures for the production of unfired granular materials, for the production of expanded granular materials [29,30,31,32,33,34,35]. A type of thermal energy waste is an aluminosilicate microsphere, a glass–ceramic material consisting of particles 20–500 µm in size [36,37]. The use of thermal energy waste in the production of lightweight concrete makes it possible to prevent negative environmental impacts and develop environmentally friendly energy-efficient technologies.

As porous fillers, glass–ceramic materials obtained using technogenic glass are widely used [38,39,40,41,42,43,44]. To obtain alkaline silicate granular materials, liquid glass is often used, which is capable of performing several functions as part of the raw mixture: binding finely dispersed components of the molding mass and formation of porosity during heat treatment of granules [45,46,47]. The physicochemical properties of liquid glass provide various methods of swelling. At a temperature of 120–500 °C liquid glass forms a solid foam with a density of 50–150 kg/m^3^. Expanded materials based on liquid glass favorably differ in high porosity.

Concrete is a composite material, the matrix of which is a binder paste. The properties of concrete depend on the adhesion of the aggregate to the binder paste. The surface roughness of the aggregate grains contributes to strengthening the bond with the concrete matrix. On the other hand, the strengthening of the interfacial transition zone between the concrete components is provided by the high adhesion of the binder to the aggregate. For the development of lightweight concrete technology, it is necessary to provide a combination of highly porous fillers of form stability with substances with pronounced binder properties.

The decrease in density and increase in the heat-shielding properties of lightweight concrete is achieved not only using highly porous aggregates, but also through the porosity of the matrix or by reducing its share in the structure of the composite material. This is reflected in the development of the technology of cellular concretes and concretes with a large pore structure [48,49,50].

Porous fillers are often characterized by the presence of non-crystalline or weakly crystalline silica (siliceous glass). This makes it possible to develop an alkali-aggregate reaction between the filler and the alkali hydroxides contained in the cement paste. Alkaline corrosion of aggregates is accompanied by swelling and cracking of cement concrete. Very topical developments aimed at increasing the resistance of artificial aggregates to alkaline corrosion [51,52,53]. To increase the durability of concrete, it is necessary to develop concretes based on cementless binders.

The high energy intensity of the production of Portland cement, which is characterized by a significant carbon footprint, necessitates the use of alternative clinker-free binders, the technological properties of which are not inferior to traditional cements. The technical achievements of recent years indicate the promise of alkaline silicate binders, the technology of which is based on the alkaline activation of aluminosilicate substances [54,55,56,57,58,59,60,61,62,63]. The impact of alkalis, for example, liquid glass, contributes to the dissolution and subsequent polycondensation of the aluminosilicate filler with the formation of a durable paste.

Liquid glass is an aqueous solution of alkali silicates, characterized by chemical activity, adjustable density, astringent properties, adhesive ability and high sensitivity to thermal effects. With numerous advantages, water glass is distinguished by its inability to bulk harden and low water resistance. The introduction of various fillers and modifiers into liquid glass makes it possible to solve these problems. The water resistance of liquid glass materials is proportional to the amount of silica gel. This is facilitated by the use of fillers containing active silica. Industrial wastes serve as an important reserve for effective fillers [18,21,26,43,44,57,58]. The variety and technological properties of liquid glass materials are provided by a combination of fillers of various origins. The unique properties of liquid glass are actively used in the technologies of porous materials [42,43,44,45,46,47]. On the basis of liquid glass, cellular concretes, granular materials and piece products are obtained. The physicochemical features of liquid glass make it possible to implement numerous variants of chemical and thermal swelling, which involves saturation of the viscous-plastic mass with the gas phase. Additive-free liquid glass is characterized by low-temperature swelling, the formation of a highly porous structure with an uneven distribution of cells of various sizes and low mechanical strength. The introduction of fillers can change the activity of the swelling process and affect the structure of the porous material. According to the nature of the effect on the thermal transformations of liquid glass, fillers are divided into inert, gel-forming and thermosetting [34,35,45]. The use of liquid glass in combination with high-performance fillers makes it possible to create materials with a rigid structure, low thermal conductivity, incombustibility, high manufacturability and environmental friendliness at a relatively low cost. For the development of effective heat-insulating liquid glass materials, the choice of fillers is important, aimed at optimizing the technological state of the compositions.

An analysis of the results of the above literature review indicates that the unique properties of liquid glass are widely used both in the technology of porous aggregates and in the technology of binders for concrete. However, there is practically no information about lightweight concretes, the filler and matrix of which are obtained from liquid glass mixtures of a related composition.

The aim of the work is to create a heat-insulating composite material based on porous granules and a matrix from mixtures of liquid glass and thermal energy waste.

To achieve this aim, the following tasks are defined:-Substantiation of the composition and processing parameters of the molding sand to obtain porous liquid glass granules and the study of the properties of the granulated material;-Study of the effect of curing temperature on the properties of the liquid glass binder;-Development of the composition and study of the properties of a composite material with a density of not more than 650 kg/m^3^ based on porous granules and a matrix of liquid glass mixtures;-Development of the fundamentals of technology for the production of granular and composite materials.

The research is based on the hypothesis that the genetic relationship of raw mixes for the matrix and filler of composite materials will ensure the reliability of the adhesion of the components and the resistance of concrete to corrosion processes, and ensure the compactness of the technological scheme of production.

## 2. Materials and Methods

### 2.1. Materials

To obtain porous granules and a composite material based on them, mixtures containing water glass and various types of thermal energy waste were used (Figure 1, Table 1).

Liquid glass (Promsteklotsentr, Yekaterinburg, Russia) is a sodium silicate aqueous solution (Na_2_O n SiO_2_ + m H_2_O). The properties of liquid glass depend on the value of the silicate modulus, which is defined as the molar ratio *n* = SiO_2_: Na_2_O. Liquid glass has high adhesion to most materials. Liquid sodium glass with *n* = 2.7 and a density of 1350 kg/m^3^ was used in the experiments.

Liquid glass served as a binder in the molding sand for the production of granules and contributed to the porosity of the granules during heat treatment. In a composite material, liquid glass is the basis of the matrix. Consequently, the multifunctionality of liquid glass has been realized in research: binding of dispersed materials; influence on the rheological properties of molding sands; activation of the participation of fillers in the processes of structure formation; thermal swelling with the formation of thin-walled cells.

As fillers for liquid glass mixtures, thermal energy wastes were used: fly ash and aluminosilicate microsphere. Fly ash and aluminosilicate microsphere are formed during the combustion of coal at a large thermal power plant in Kazakhstan (Ekibastuz, Pavlodar region).

Fly ash from thermal power plants is a finely dispersed powder formed during the combustion of pulverized coal. The structure of fly ash particles is due to the short-term stay of the fuel in the high temperature zone. Fly ash particles are rounded fused grains, many of which have the smallest, mostly closed pores. The size of spherical particles of fly ash is 0.5–100 µm. Fly ash consists of aluminosilicate opaque glass, contains particles of quartz, mullite, unburned coal. The specific surface area of the fly ash is 280–300 m^2^/kg. Bulk density is 670–720 kg/m^3^. The use of fly ash in the composition of liquid glass compositions helps to reduce thermal conductivity and increase the fire resistance of the material.

Aluminosilicate microsphere is a light fraction of ash, which is formed during high-temperature flaring of coal. Aluminosilicate microsphere is a bulk material consisting of glass–ceramic hollow particles of spherical shape with a diameter of 50–200 µm. Bulk density 380–400 kg/m^3^. The use of microspheres as a filler is due to a number of advantages: high dispersion contributes to the creation of homogeneous structures, low density, high strength, increased thermal stability and resistance in aggressive environments.

The content of the main fractions in the waste from the thermal power industry is shown in Figure 2.

Humidity of fly ash and aluminosilicate microspheres sent by the supplier should be no more than 1% according to the Russian standard GOST 25818-2017. Humidity of thermal power waste used in experiments was determined by drying samples of materials to constant weight at a temperature of 110 °C. In the experiments, it used thermal energy waste with a natural moisture content of 0.5–0.8 wt.%.

### 2.2. Mix Design

To determine the range of compositions of the molding masses that provide the production of granules and a binder of composite materials, water–glass mixtures that differ in the composition and content of the filler were studied (Table 2). The mixtures were prepared by thorough mixing of the components. Changes in the material composition led to significant differences in the state of liquid glass mixtures. To characterize liquid glass mixtures, a visual assessment of their molding properties was used (Table 3). The values of a “liquid glass: filler” ratio (hereinafter “LG:F”) are taken taking into account preliminary experiments. At values of “LG:F” less than 0.65, friable mixtures are formed, which require intensive compaction during molding. An increase in the value of “LG:F” over 1.25 is accompanied by the formation of a very mobile suspension, prone to delamination.

Comparison of the characteristics of the studied mixtures made it possible to reveal the following regularities. With an increase in the content of aluminosilicate microspheres in excess of 20% in the composition of the filler, the plastic properties of liquid glass mixtures decrease. A filler containing more than 80% aluminosilicate microspheres increases the thixotropy of the mixtures and makes it difficult to control the molding properties, therefore it is considered inappropriate. The use of a combined filler containing 40–80% aluminosilicate microspheres makes it possible to reduce the density of liquid glass mixtures. This is relevant for obtaining porous materials.

Liquid glass mixtures are highly viscous masses, the properties of which depend on the type and content of the filler. The introduction of fillers increases the viscosity of the liquid glass mixture due to polycondensation processes. It was noted [64] that in the presence of fillers with a high content of silica, the increase in the viscosity of liquid glass mixtures occurs more intensively. From the standpoint of ionic theory, this behavior of liquid glass masses is explained by the complexing properties of silica, which contributes to an increase in the silicate modulus of the mixture. This statement is consistent with the nature of the change in the state of the investigated liquid glass mixtures, which largely depends on the content of aluminosilicate microspheres.

Liquid glass mixtures, designated as “plastic” and “very plastic”, are characterized by pronounced cohesion and exhibit a tendency to pelletization. Such mixtures are preferred for the production of granular material. Taking into account the influence of aluminosilicate microspheres on the density and molding properties of mixtures, compositions 9, 10, 11, 15, 16 and 17 were used for granulation.

To obtain a heat-insulating composite material based on a porous filler, it is advisable to form a large-pore structure with a minimum proportion of a binder. Contact monolithic porous granules require a binder with high adhesion.

Analysis of the research results (Table 2 and Table 3) shows that mixtures 31, 32, 33, 37, 38, 39 and 40 exhibit adhesive ability. Liquid glass mixtures 33, 39 and 40, including 40–60% aluminosilicate microspheres, are recognized as preferable.

### 2.3. Methods

#### 2.3.1. Preparation of Raw Materials

Waste from thermal power engineering was sifted through a sieve with a mesh size of 0.315 mm to separate foreign particles. The dispersity of thermal power generation wastes was assessed using an FSH-6K photosedimentometer (Pribory Khodakova, Moscow, Russia). The density of liquid glass was determined with a hydrometer.

#### 2.3.2. Molding and Research of Properties of Granules

Figure 3 presents a flowchart of research of properties of granules.

A thoroughly mixed mixture of dry components was loaded into a drum granulator, into which liquid glass was introduced by irrigation. The duration of pelletizing in the granulator was 5–10 min, taking into account the diameter of the resulting granules. To assess the structural and mechanical properties of molding sands, the plastic strength was determined using a conical rheometer and precision balance. The crushing load of the granules evaluated the strength of the molded (raw) granules. The breaking load corresponded to the mass of the load at which cracks formed in the granule. The molded granules were subjected to heat treatment in an oven at a given temperature. The microstructure of the materials was studied using a JSM-649OLV energy scanning electron microscope (JEOL, Tokyo, Japan).

#### 2.3.3. Preparation and Study of Liquid Glass Binder

Figure 4 presents a flowchart of research of liquid glass binder. 

Fly ash and aluminosilicate microspheres were successively introduced into liquid glass and thoroughly mixed until a homogeneous suspension was formed.

The condition of the binder suspension was assessed by the diameter of the slump flow on a Suttard viscometer. Samples of the liquid glass binder were subjected to heat treatment in an oven. Samples of binder paste were tested for strength and water resistance. To determine the phase composition of the binder, a modernized DRON-3M diffractometer (Burevestnik, Moscow, Russia) was used. The diffractometer was equipped with a BSV-24 X-ray tube with α-CuK radiation. The XRD patterns were processed using the difWin software.

#### 2.3.4. Preparation of Molding Mixes and Production of Samples of Composite Materials

Figure 5 presents a flowchart of research of composite materials.

Fly ash and aluminosilicate microspheres were poured into liquid glass, and stirred for 2 min. A porous granular filler was loaded into the resulting suspension and mixed for 3 min to evenly distribute the binder between the granules. The flowability of the mixes was evaluated using the Abrams cone. In the experiments, molding sands with a cone slump of 2–4 cm were used. The molding mixes were placed in a metal mold and vibrated for 30 s. Samples after preliminary exposure for 1 h were subjected to heat treatment in a drying chamber at a given temperature.

#### 2.3.5. The Tests

To determine the properties of the granular material, test methods were used according to the Russian standards GOST 9758-2012 and GOST 32496-2013. When determining the properties of the composite material, methods were used according to the Russian standard GOST 25820-2014.

The total porosity (P, %) of fired granules was determined taking into account the true density of the substance of the granules (ρ_t_, kg/m^3^) and the average density of the granules (ρ_a_, kg/m^3^).
$$P=\frac{\rho_{t}-\rho_{a}}{\rho_{t}}\;100$$

The true density of the substance of the granules was determined by the psychometric method. The average density of the granule (ρ_a_, kg/m^3^) was determined as the ratio of the mass (m, kg) to the volume of the granule, for the calculation of which the granule diameter (D, m) was determined:$$\rho_{a}=\frac{m}{\frac{\pi \cdot D^{3}}{6}}$$

The strength of the fired granules was determined by the splitting method. The strength of each granule during splitting (R_g_, MPa) was calculated taking into account the maximum splitting force (G, N) and the split area of the granule (F, m^2^):$$R_{g}\;=\;\frac{G}{F}$$

The granule splitting strength was determined as the arithmetic mean of the results of 10 tests.

Water absorption (W,%) of the granules was determined taking into account the mass of the initial sample (m, g) and the mass of the sample saturated with water (m_w_, g) for a given time:$$W=\frac{m_{w}-m}{m}\;100$$

The samples were tested for strength by a PGM-1000MG4 hydraulic press (Stroypribor, Moscow, Russia). An ITP-MG4 instrument (Stroypribor, Moscow, Russia) measured the thermal conductivity coefficient of composite materials.

The water resistance of composite materials was evaluated by the softening coefficient (K_w_), taking into account the strength of the sample that was in water (R_w_, MPa) for a given time, and the strength of the original sample (R, MPa).
$$\mathrm{Kw}=\frac{R_{w}}{R}$$

The resistance to frost destruction of composite materials was determined by comparing the strength of samples subjected to alternate freezing (temperature “minus” 20 °C) and thawing in water (temperature 20 °C) after 25 and 50 test cycles. Test cycle: freezing during 4 h and thawing in water during 4 h.

The heat resistance test of composite materials was determined by the ability of the samples to withstand sudden temperature changes. The samples were placed in a drying chamber preheated to a temperature of 250 °C and kept for 2 h. Then the heated samples were immersed in water (temperature 20 °C) for 2 h exceeded 20% of the initial weight of the samples.

The heat resistance test of composite materials was determined by the ability of the samples to withstand sudden temperature changes. The samples were placed in a drying chamber preheated to a temperature of 250 °C and kept for 2 h. Then the heated samples were immersed in water (temperature 20 °C) for 2 h. Thermal stability was determined by the number of heat cycles in which the samples retained their integrity, and the weight loss did not exceed 20% of the initial weight of the samples.

The heat resistance of composite materials was determined by the state of appearance and “residual” strength. The tests were carried out into a muffle furnace at a temperature of 1050 °C, where the samples were kept for 4 h. After turning off the furnace, the samples were kept at a temperature of 500 °C for 2 h, then again subjected to high-temperature exposure. For heat-resistant materials, the “residual” strength should be at least 80% of the strength of the specimens not tested.

## 3. Results and Discussion

### 3.1. Influence of Technological Factors on the Formation and Properties of Liquid Glass Granules

Porous granular aggregate is the main component of lightweight concrete. The structural characteristics of the granules affect the thermal and physical-mechanical properties of concrete.

The properties of granules obtained from molding sands (Table 4), the composition of which was previously determined in Section 2.2 (Table 2), were investigated. The plastic strength of the molding sands was determined after 15 min from the moment of preparation. The values of plastic strength make it possible to predict the molding properties of the mixture during the granulation period. Preference is given to mixtures with increased values of plastic strength.

The density of molded (raw) granules depends on the material composition of the mixture, the size of the granules and serves as a preliminary criterion for the density of the fired granules. Wet granules with an average diameter of 10.5 mm were used to determine the density.

The shaped granules must have mechanical strength in order to maintain their integrity during transportation to the thermal plant. The loads experienced by green granules during transport are modeled by crush and drop tests. The crush test was carried out on pellets with an average diameter of 10.5 mm. The granules obtained from mixtures with a lower content of liquid glass achieved increased strength and, when dropped from a height of 300 mm, withstood at least 12 drops.

Based on the totality of the characteristics of the molded granules, mixtures 9 and 1011, which differ in the composition of the filler, were selected for further research. Granules with a diameter of 10–11 mm were fired at temperatures of 150, 200, 250 and 300350 °C with an isothermal exposure of 40 min. The heating rate of the drying cabinet is 10 °C/min. The firing temperature was set taking into account pyrogenic transformations in liquid glass and compositions based on it [64,65].

An increase in the firing temperature is accompanied by a decrease in the density and water absorption of the granules (Figure 6 and Figure 7). At the same time, the density of granules from mixtures containing 30 and 40% aluminosilicate microspheres reaches the lowest values (Figure 6). However, the granules from the mixture with the highest content of aluminosilicate microspheres collapsed after 2 days of exposure to water.

For a detailed study of the properties of the granular material, mixture 10 (Table 4) containing 30% aluminosilicate microspheres was used (Table 5).

The bulk density of the granular material depends on the size of the granules and decreases with increasing firing temperature (Table 5).

Porization of liquid glass materials is carried out by thermal swelling, due to pore-forming components. During the firing of materials, liquid glass serves as a source of the gas phase, which ensures the porization of the pyroplastic mass. Heating of liquid glass is accompanied by a decrease in mass by 50–55%. The porization of liquid glass proceeds in several stages, the formation of pores depends on the type and amount of water contained in the material. At temperatures up to 100–120 °C, free and adsorbed water evaporates from liquid glass, partial porization and material hardening occurs [64,65,66,67,68,69].

Firing in the temperature range of 130–180 °C promotes the removal of crystallization water and secondary pore formation in the material. Removal of hydration water from liquid glass begins at a temperature of 240–245 °C and ends at 600–650 °C. Therefore, the formation of material porosity is determined by adsorption and crystallization water. Fillers reduce the porosity of liquid glass, provide a more uniform distribution of pores.

The formation of pores in the studied granules occurs with the participation of liquid glass and due to the aluminosilicate microsphere. The porosity of liquid glass granules varies slightly in the studied temperature range (Table 6). The main part of the pores in the granules is formed at a temperature of 150 °C. The expansion of the mixture due to liquid glass provides the granules with the correct spherical shape. At the same time, the volume of granules increases by 1.08–1.15 times. Hollow particles of aluminosilicate microspheres are the carrier of porosity and ensure the stability of the structure of swollen samples, performing a frame-forming function.

The study of the structure of the granules indicates a uniform distribution of porosity (Table 5). A significant part of the pore space of the granules is formed from isolated volumes of hollow particles of aluminosilicate microspheres with a diameter of 50–200 µm (Figure 8). Particles of aluminosilicate microspheres form the basis of the contact structure. The other part of the pores is formed in the liquid glass mass filled with fly ash, and is represented by closed cells, the average size of which is 1–10 µm. The cellular liquid glass mass envelops and holds together the microspheres, fills the space between the particles. The granules have voids 10–100 µm in length, not filled with a substance. The volume of voids is on average 10–20% and decreases with increasing firing temperature (Figure 8). Water absorption of granules is mainly due to the presence of voids (Figure 7). The study of the microstructure was carried out on the surface of the split granules. The splitting of the granules is accompanied by the destruction of hollow particles, which indicates a high adhesion of the liquid glass substance to the aluminosilicate microsphere.

The physical and mechanical properties of the granules significantly depend on the firing temperature. An increase in the strength and water resistance of granules with an increase in temperature above 150 °C indicates transformations in the composition of the liquid glass mass filled with fly ash. During thermal porization of the liquid glass mixture, solid solutions are formed with the participation of fillers, partial crystallization of silicate new growths occurs.

### 3.2. Study of the Binder for Composite Materials

Composite materials contain a matrix and a discrete component (filler). The matrix provides a uniform distribution of stresses throughout the volume of the material, protects the discrete component from environmental influences, and determines the properties of the composite.

For the development of composite materials, the use of a liquid glass matrix is provided such a suspension of liquid glass and filler. The filler regulates the rheological properties of the liquid glass binder, participates in the structure formation of the alkali silicate paste.

Liquid glass compares favorably with high adhesion to materials of various chemical nature, incombustibility and non-toxicity. The disadvantages of liquid glass include the inability to bulk hardening and low water resistance of composite materials. To cure liquid glass, modifiers (hardeners) of various compositions and (or) heat treatment are used [52,56,61,70]. To obtain concrete with a cellular structure, pore-forming components are introduced [42,44,48,50]. Indicators of strength and water resistance of liquid glass materials are proportional to the content of slightly soluble silica gel, which is the basis of the adhesive. Silicon-containing additives contribute to the polymerization of liquid glass components. Chemically active compounds are often used as silicon-containing substances. In the technology of liquid glass materials, sodium fluoride silicate Na_2_SiF_6_ is widely used, which provides intensive curing of the binder.

However, the use of hardeners in liquid glass mixtures often creates technological difficulties associated with the regulation of molding properties; does not exclude the use of thermal effects on the hardening mass. Heat treatment in a humid environment worsens the heat-shielding properties of porous materials. A number of hardeners are scarce substances, and sodium fluoride silicate belongs to dangerous reagents.

As a matrix of the developed composite materials, it is proposed to use liquid glass mixtures 33, 39 and 40 (Table 2), which differ in the ratio “LG:F”.

The creation of a composite material with a large-pore structure requires the use of a matrix that is easily distributed between the aggregate grains and forms thin shells of adhesive on its particles.

Model masses from liquid glass mixtures 33, 39 and 40 and developed granules were studied (Section 3.1). The model mass of the liquid glass mixture 39 compares favorably with a uniform distribution over the surface of the granules with the formation of a narrow contact zone between the particles (Table 7). This will minimize the content of the matrix in the structure of the composite material.

Liquid mixtures for molding granules (Table 4) and matrix substance (Table 7) contain the same components in different ratios. An analysis of the research results (Section 3.1) made it possible to assume that the nature of the effect of heat treatment of a liquid glass binder will be similar to the effect of firing on granules.

Samples 40 × 40 × 40 mm in size, made from a liquid glass binder composition 39, after a preliminary exposure for 1 h, were subjected to heat treatment at various temperatures with an isothermal exposure for 2 h.

Heat treatment of the liquid glass binder ensures the formation of a stone of a porous structure (Table 8). The porosity of the fired binder is on average 52–60%. The nature of pore formation in the binder and granules is similar. The lower porosity of the binder compared to the granules is due to the reduction in the aluminosilicate microsphere in the liquid glass mixture. The structure of the fired binder is characterized by a high content of porous liquid glass mass (Figure 9). The effect of the firing temperature on the physical and mechanical properties of the binder is significant in the range of 150–250 °C. An increase in temperature above 150 °C is accompanied by a decrease in density and an increase in the water resistance of the binder stone. The properties of the binder fired at temperatures of 250, 300 and 350 °C are characterized by similar values. This can serve as a rationale for clarifying the technological regime of heat treatment in the direction of lowering the temperature.

The fired liquid glass mass consists of crystalline and X-ray amorphous phases (Figure 10). The presence of crystalline phases of β-quartz, α-crystobalite and wollastonite is mainly due to the participation of fly ash in the processes of structure formation of the binder paste. An increase in the crystalline component in the binder with an increase in the firing temperature contributes to the hardening and increase in the water resistance of the paste (Figure 11, Table 8).

The results of the studies carried out testify to the expediency of heat treatment of a liquid glass binder filled with waste from thermal power engineering at a temperature of 250–350 °C. Firing of the liquid glass binder ensures the formation of a low-density waterproof paste.

### 3.3. Research of Liquid Glass Composite Materials

The properties of composite materials are largely determined by the ratio of the components of the molding mix. Composite materials obtained from molding mix with different filler content have been studied. The compositions of the molding mixes were evaluated by the filling factor C_f_ (the ratio “volume of binder: volume of filler”). Molding mixtures were prepared on the basis of liquid glass binder composition 39 (Table 7). Granules 7–10 mm in size were used as filler (Table 5). The flowability of the molding mixes corresponded to a slump of 2–4 cm. Molded samples 70 × 70 × 70 mm in size were subjected to heat treatment in an oven according to the regime: 1 h—holding at room temperature, 1.5 h—heating to a temperature of 350 °C, 2.5 h—isothermal exposure, 1 h—cooling to a temperature of 150–180 °C.

Composite materials characterized by C_f_ = 0.23–0.33 have a large-pore structure. Porous granules coated with a thin shell of a binder form a monolith due to contact at the points of contact (Figure 12). Composite materials characterized by C_f_ = 0.63–0.73 have a continuous structure; the matrix fills the intergranular space and covers the granules (Figure 12). In composite materials with C_f_ = 0.43–0.53, the matrix partially fills the space between the grains with filler. The nature of the split of samples with C_f_ = 0.63–0.73 indicates the reliability of adhesion of the porous filler to the matrix.

Changing the ratio between the components of the molding sand allows to obtain composite materials of various structures with a range of density changes of 380–650 kg/m^3^ and strength of 3.5–9.0 MPa (Figure 13).

The composite material with C_f_ = 0.43 was studied for resistance to various environmental influences. In the manufacture of samples, a technique used in the technology of lightweight concrete products was used: the formation of a three-layer structure. The outer layers of the product, made from a fine-grained mixture, provide increased clarity of the geometric dimensions of the product and perform protective functions [71,72,73,74]. To form the outer layers, a molding mix containing a liquid glass binder (mixture 39) and crushed particles of a porous filler was used. The choice of three-layer test specimens not only allows one to simulate the behavior of products, but also to evaluate the reliability of adhesion of various layers of a composite material. To enhance the effect of test conditions on the material, samples with cut ends were used, on which access to the inside of the granular aggregate was torn off. The main characteristics of the liquid glass composite material are shown in Table 9.

The developed composite material showed resistance to water, frost aggression and variable effects of elevated and high temperatures. The nature of the fractures of the samples during strength testing after various impacts indicates a high adhesion of the binder and filler (Figure 14a), reliable adhesion of words of various structures (Figure 14b). Signs of destruction of the structure under conditions of variable temperatures appear mainly in areas with defects in the formation of samples (Figure 14c,d). The samples subjected to the heat resistance test are characterized by shrinkage of 2–3%, an increase in density by 15–17%, and hardening.

Therefore, the developed liquid glass composite material, which is a kind of cementless lightweight concrete, is characterized by a strong porous structure, has heat-shielding properties and is resistant to various operational factors.

### 3.4. Fundamentals of the Technology of Liquid Glass Composite Material

The developed porous granular material can be used as a heat-insulating backfill, and also serve as a filler in lightweight concrete.

The proposed composition and method for curing the liquid glass binder can be implemented in the technology of composite materials of various structures.

The expediency of combining these materials is confirmed by the results of this study. Taking into account the commonality of raw materials and the nature of technological processing, a basic technological scheme for obtaining a liquid glass composite material has been developed (Figure 15).

The minimum list of raw materials that do not require pre-treatment ensures the compactness of the technological scheme. In the joint production of granular filler and composite material, it is possible to reduce the firing temperature of the granules to 150–200 °C. The strength of the granules of low-temperature firing is sufficient to maintain their integrity during further technological processing. The firing of the composite material at a temperature of 300–350 °C will ensure the curing of the matrix and the strengthening of the porous filler. The results of laboratory tests indicate the comparability of the properties of composite materials obtained using granules of different processing.

## 4. Conclusions

Ideas about the technology of liquid glass compositions based on thermal energy waste have been developed.

A composite material has been developed, the matrix and porous filler of which are synthesized from mixtures of liquid glass, fly ash and aluminosilicate microspheres. A directed change in the material composition of liquid glass mixtures provides the specified technological properties of the raw mass. Thermal swelling of liquid glass and the presence of hollow particles of aluminosilicate microspheres provide the formation of the cellular structure of the granules and the matrix substance.

Controlling the content of the binder and porous filler makes it possible to obtain a heat-insulating liquid glass material with a thermal conductivity of 0.095–0.100 W/(m·°C), a density of 380–650 kg/m^3^ a strength of 3.5–9.0 MPa. The genetic commonality of the matrix and porous granules guarantees reliable adhesion of the components. The presence of an insoluble crystalline phase and predominantly closed porosity of the components ensure the stability of the composite material under conditions of variable values of humidity and temperature.

The developed liquid glass granular material with a diameter of 7–20 mm, obtained by firing at a temperature of 250–350 °C, is characterized by a porosity of 76–78%, a bulk density of 175–265 kg/m^3^ a grain strength of 2.4–3.3 MPa.

For curing a composite material based on a liquid glass binder, heat treatment at a temperature of 300–350 °C is proposed, which ensures the formation of a porous waterproof matrix with high adhesion to the filler. The proposed curing conditions exclude the use of special hardeners and pore-forming components, promote the adhesion of layers in the material of the combined structure.

A basic technological scheme has been developed that provides for the joint production of granular and composite materials and allows to reduce the temperature at the stage of granule firing to 150–200 °C.

The resource saving of the developed composite material technology is ensured by the use of a cement-free binder with a low carbon footprint, the involvement of large-tonnage waste from thermal power engineering, and low-temperature technological processes.

## Figures and Tables

**Figure 1 polymers-14-03461-f001:**
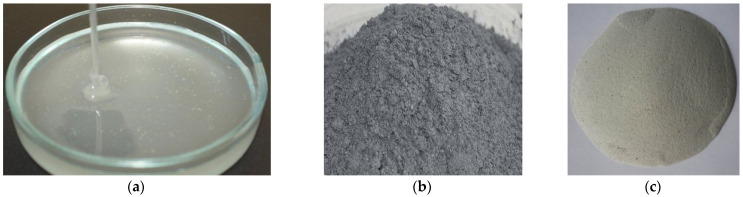
Appearance of raw materials used: (**a**) liquid glass; (**b**) fly ash, (**c**) aluminosilicate microsphere.

**Figure 2 polymers-14-03461-f002:**
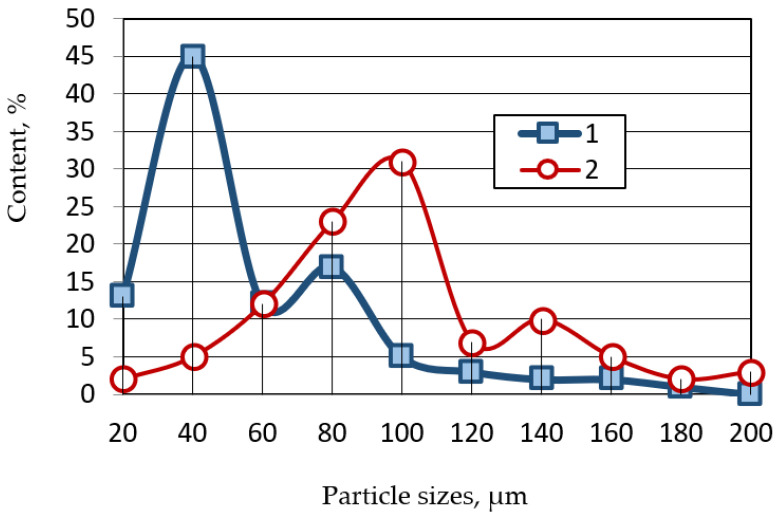
Fractional composition of thermal energy waste: 1—fly ash; 2—aluminosilicate microsphere.

**Figure 3 polymers-14-03461-f003:**
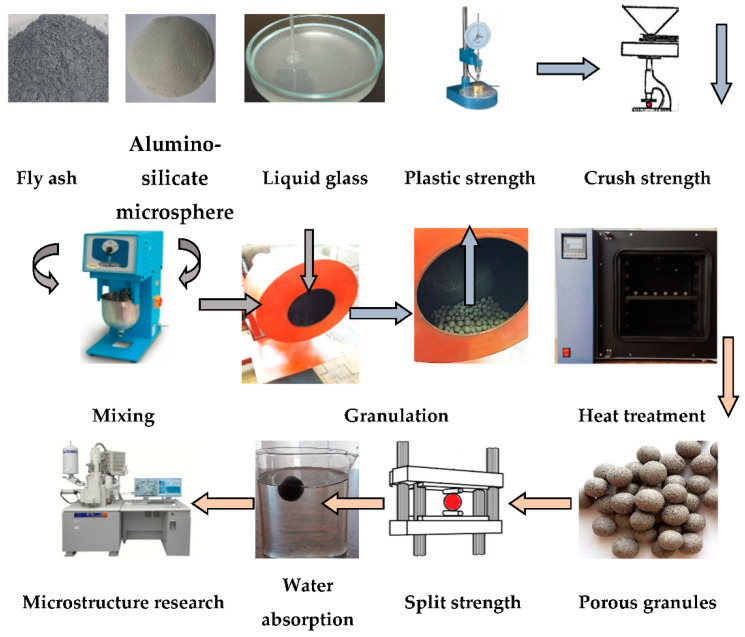
Flowchart of research of properties of granules.

**Figure 4 polymers-14-03461-f004:**
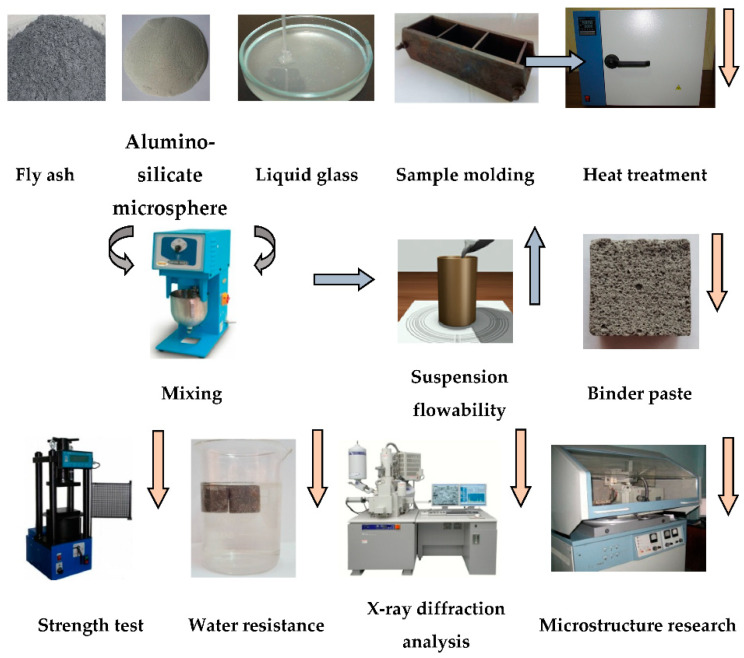
Flowchart of research of liquid glass binder.

**Figure 5 polymers-14-03461-f005:**
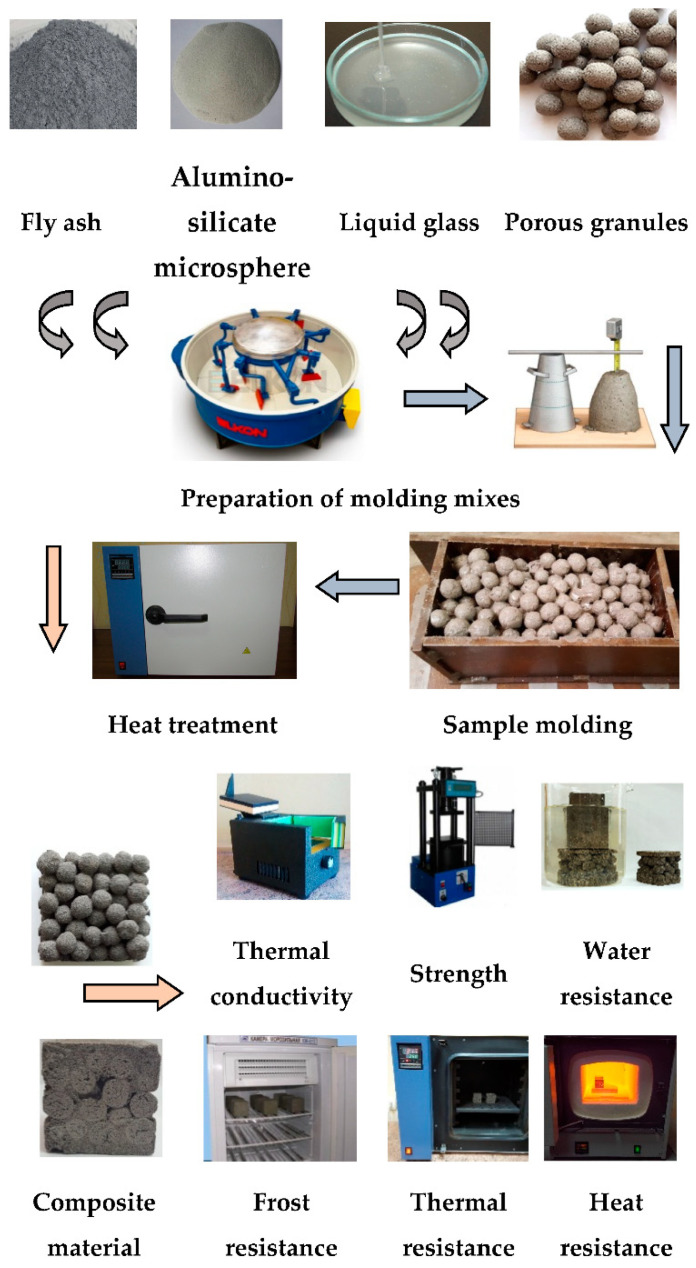
Flowchart of research of composite materials.

**Figure 6 polymers-14-03461-f006:**
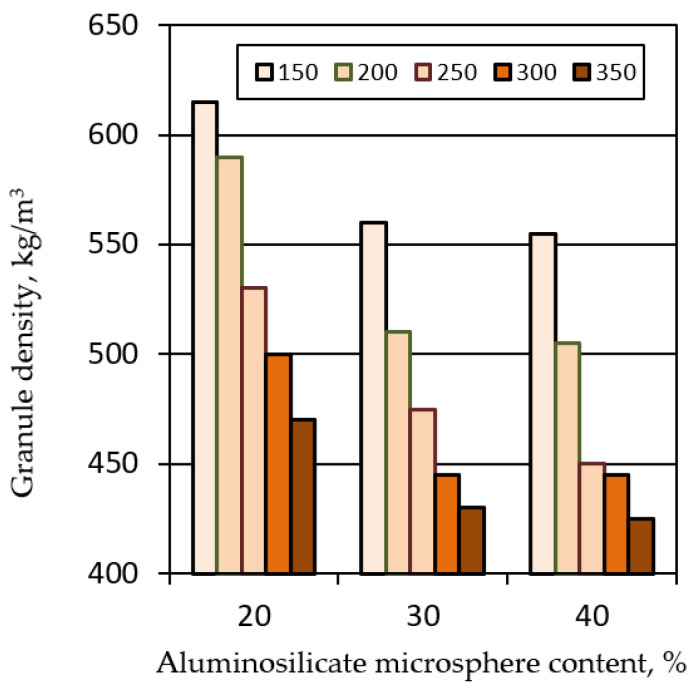
Effect of firing temperature and filler composition on the density of liquid glass granules.

**Figure 7 polymers-14-03461-f007:**
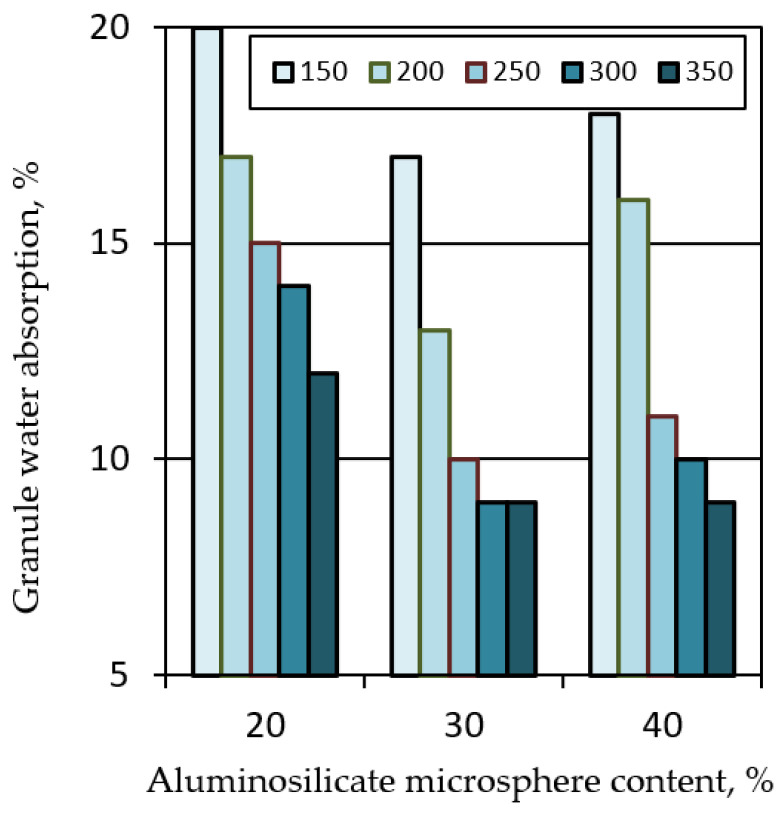
Effect of firing temperature and filler composition on the water absorption of liquid glass granules.

**Figure 8 polymers-14-03461-f008:**
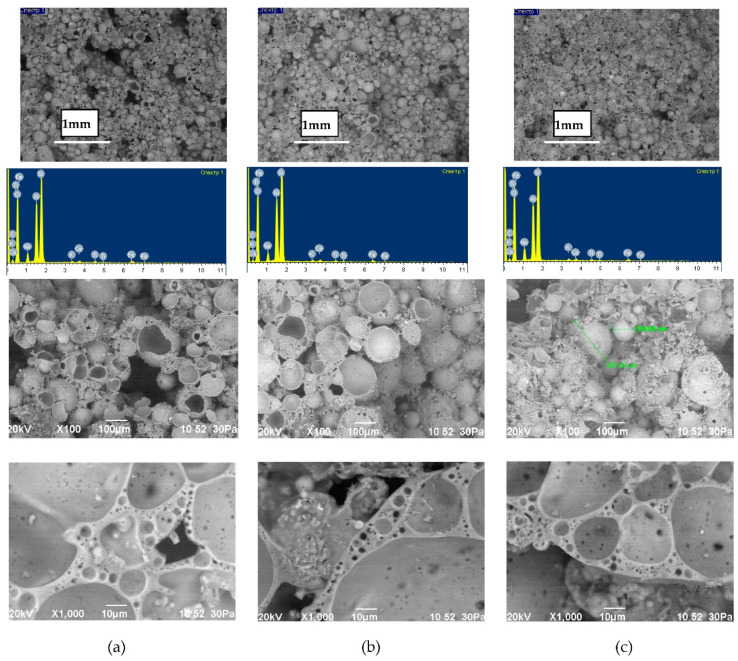
Microstructure of granules from a mixture of 10, fired at different temperatures, °C: (**a**) 150; (**b**) 250; (**c**) 350.

**Figure 9 polymers-14-03461-f009:**
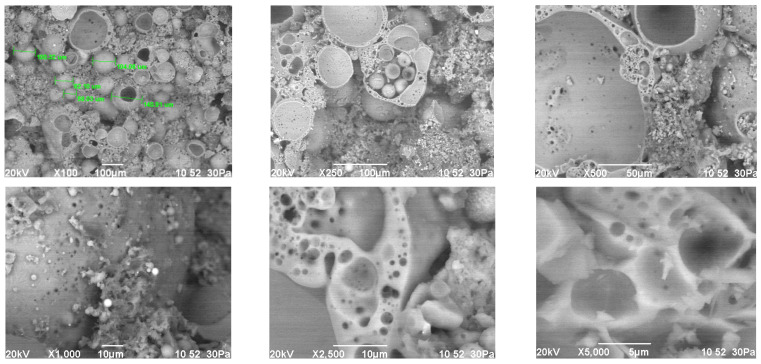
The microstructure of the liquid glass binder (mixture 39), fired at 350 °C.

**Figure 10 polymers-14-03461-f010:**
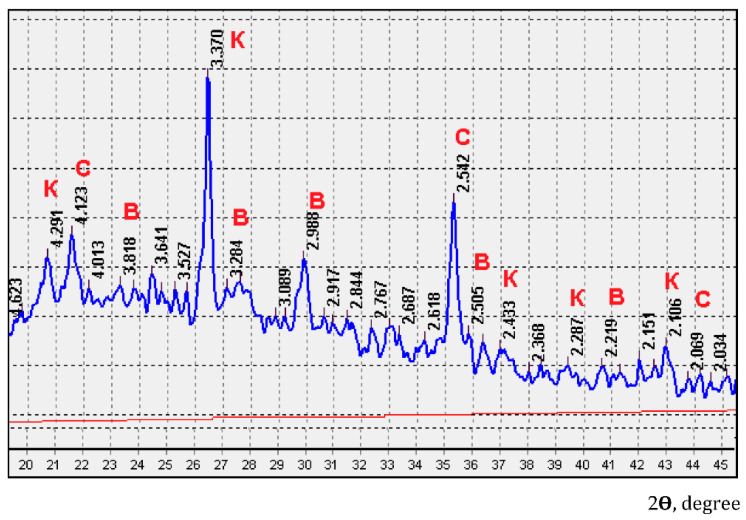
X-ray diffraction pattern of the liquid glass binder, fired at temperature of 350 °C: K—β-quartz; C—α-cristobalite; B—wollastonite.

**Figure 11 polymers-14-03461-f011:**
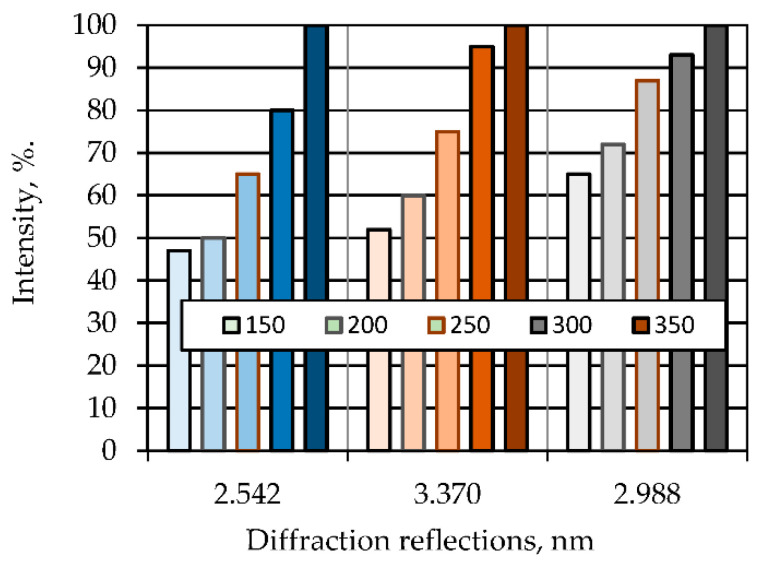
Effect of firing temperature on the content of crystalline phases in the liquid glass binder: 2.542 nm-α-crystobalite; 3.370 nm-β-quartz; 2.988 nm-wollastonite.

**Figure 12 polymers-14-03461-f012:**

The structure of liquid glass composite materials with different ratio “binder volume: filler volume”.

**Figure 13 polymers-14-03461-f013:**
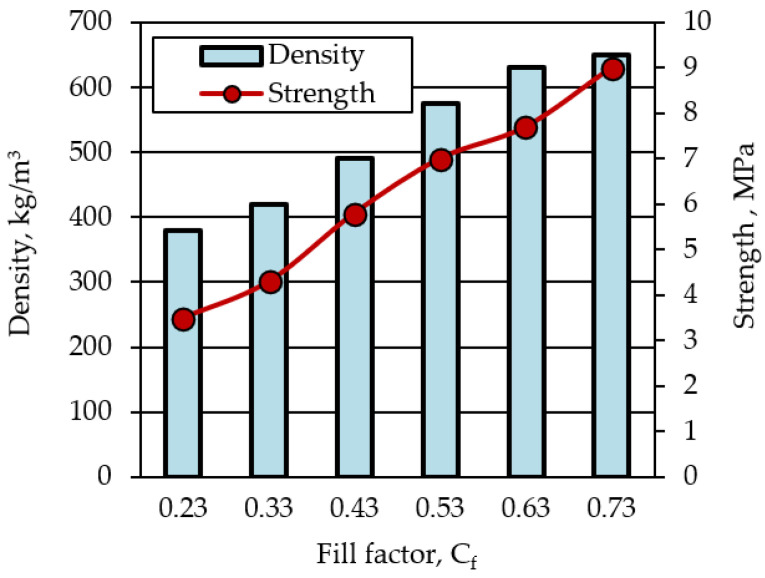
The influence of the fill factor of liquid glass mixtures on the properties of composite materials.

**Figure 14 polymers-14-03461-f014:**
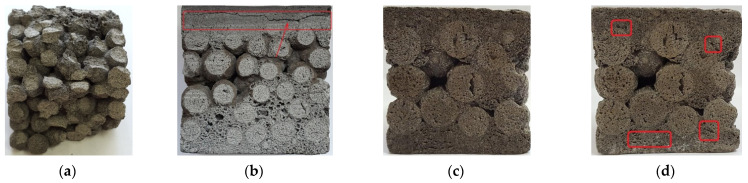
Appearance of composite material samples after testing: (**a**) after compression test; (**b**) destruction of a multilayer sample; (**c**) before testing for thermal resistance; (**d**) after 25 thermal cycles of thermal resistance tests.

**Figure 15 polymers-14-03461-f015:**
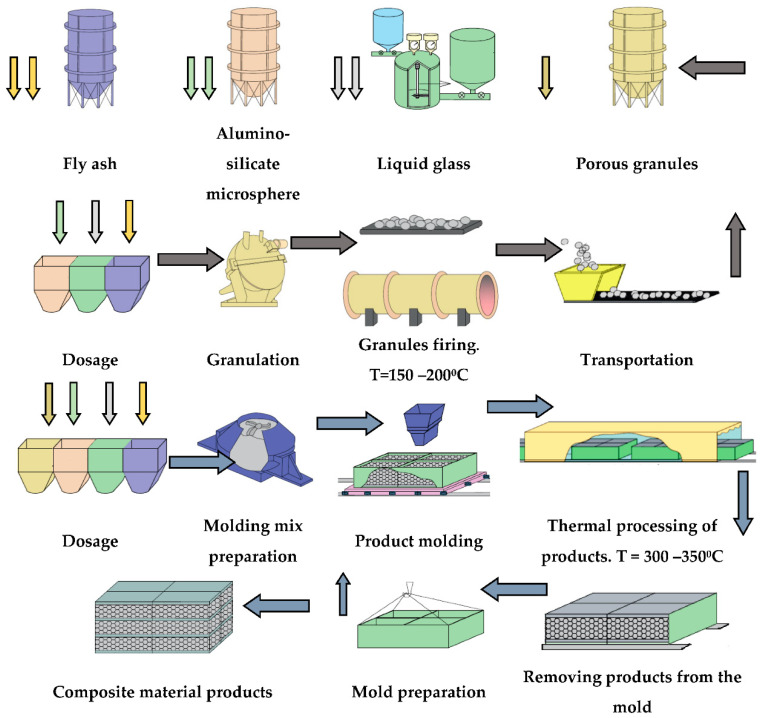
Principal technological scheme of joint production of porous granules and composite material from liquid glass mixtures.

**Table 1 polymers-14-03461-t001:** Chemical composition of the feedstock.

Content of Oxides, %	Liquid Glass	Fly Ash	**Aluminosilicate Microsphere**
SiO_2_	28.9	48.5	68.3
Al_2_O_3_	0.1	17.8	25.5
Fe_2_O_3_	0.1	1.4	1.5
CaO	0.2	14.8	2.2
MgO	–	3.3	1.5
Na_2_O	10.7	1.3	0.3
SO_3_	–	2.6	0.7
loss on ignition	60.0	10.0	–

**Table 2 polymers-14-03461-t002:** Composition and characteristics of liquid glass mixtures.

Mix ID	Composition and Characteristics of Liquid Glass Mixtures, %		Liquid Glass: Filler	Characteristics of the Liquid Glass Mixture
	Fly Ash	Aluminosilicate Microsphere		
1	100	0	0.65	plastic
2	80	20		plastic
3	60	40		tough
4	40	60		tough
5	20	80		very tough
6	0	100		no connectivity
7	100	0	0.75	very plastic
8	80	20		very plastic
9	60	40		plastic
10	40	60		plastic
11	20	80		tough
12	0	100		very tough
13	100	0	0.85	viscoplastic
14	80	20		viscoplastic
15	60	40		very plastic
16	40	60		plastic
17	20	80		plastic
18	0	100		tough
19	100	0	0.95	viscous
20	80	20		viscous
21	60	40		viscoplastic
22	40	60		viscoplastic
23	20	80		plastic
24	0	100		plastic
25	100	0	1.05	viscous
26	80	20		viscous
27	60	40		very viscous
28	40	60		very viscous
29	20	80		viscoplastic
30	0	100		viscoplastic
31	100	0	1.15	fluid
32	80	20		fluid
33	60	40		fluid
34	40	60		viscous
35	20	80		very viscous
36	0	100		viscoplastic
37	100	0	1.25	very fluid
38	80	20		very fluid
39	60	40		very fluid
40	40	60		fluid
41	20	80		viscous
42	0	100		very viscous

**Table 3 polymers-14-03461-t003:** Gradation of the state of liquid glass mixtures.

Characteristics of the Liquid Glass Mixture	Description of the State of the Liquid Glass Mixture
No connectivity	The mixture is loose and friable. The lack of liquid glass excludes the formation of solid shells on the surface of the filler particles.
Very tough	The mixture is loose. The shells of liquid glass on the surface of the filler particles are very thin, the bond between the particles is weak.
Tough	The loose mixture consists of individual aggregates of filler particles held together by shells of liquid glass.
Plastic	The mixture is characterized by the continuity of the structure. The filler particles are densely packed and hold the liquid glass shells.
Very plastic	The mixture is characterized by the continuity of the structure. Packing of filler particles in liquid glass is loose.
Viscoplastic	The mixture is characterized by the continuity of the structure. Filler particles move freely in liquid glass.
Very viscous	The mixture sticks to the surface of the bodies.
Viscous	The mixture is distributed over the surface of the bodies.
Fluid	The mixture flows down the vertical surface of the bodies.
Very fluid	The mixture easily flows over the vertical surface of the bodies, forming a thin layer.

**Table 4 polymers-14-03461-t004:** Composition and properties of molded granules.

Mix ID *	The Composition of the Molding Liquid Mixture, wt.%			Plastic Strength of the Molding Mixes, MPa	Properties of Molded Granules	
	Fly Ash	Alumino- Silicate Microsphere	Liquid Glass		Density, kg/m^3^	Crush Strength, N/Granule
9	30	20	50	0.23	985	15.0
10	20	30	50	0.21	980	15.7
11	10	40	50	0.25	965	16.1
15	28	19	53	0.13	890	11.2
16	19	28	53	0.15	870	12.0
17	10	37	53	0.17	855	13.4

* Designation of mixtures as in Table 2.

**Table 5 polymers-14-03461-t005:** Influence of firing temperature on the properties of granules of different sizes (mixture 10).

Temperature, °C	Bulk Density, kg/m^3^, of Granule with Diameter, mm			
	7	10	15	20
150	310	270	270	245
200	380	250	245	220
250	265	250	235	205
300	255	245	210	190
350	240	230	210	175
**Appearance and macrostructure of granules**				
350	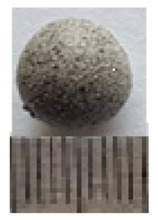	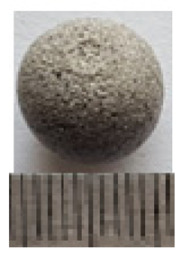	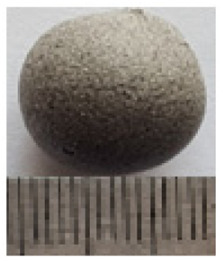	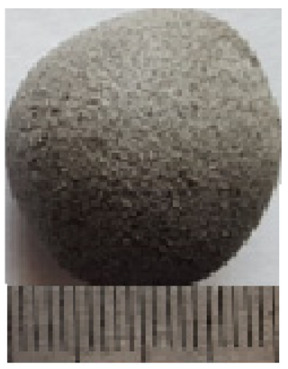
	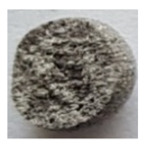	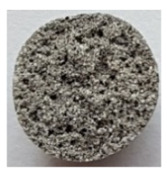	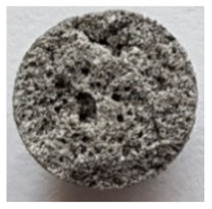	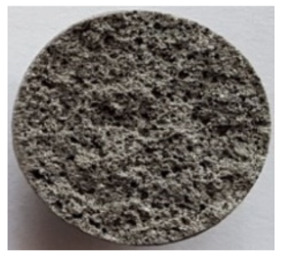

**Table 6 polymers-14-03461-t006:** Effect of firing temperature on the properties of granules with a diameter of 10 mm.

Temperature, °C	Porosity, %	Strength, MPa	Softening Co Efficient
150	73.2	1.9	0.25
200	75.2	2.5	0.70
250	76.0	3.3	0.75
300	77.5	2.9	0.81
350	78.1	2.4	0.92

**Table 7 polymers-14-03461-t007:** Composition and properties of liquid glass binders.

Mix ID *	The Composition of the Molding Liquid Mixture, wt.%			Slump Flow, mm	Appearance of the Mass
	Fly Ash	Aluminosilicate Microsphere	Liquid Glass		
33	23	16	61	150	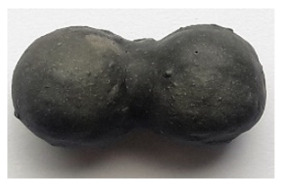
39	22	15	63	175	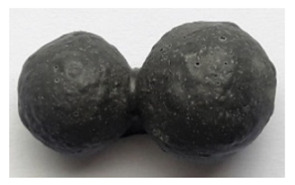
40	15	22	63	163	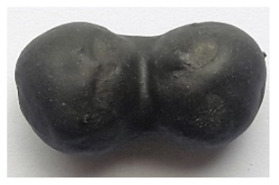

* Designation of mixtures as in Table 2.

**Table 8 polymers-14-03461-t008:** Influence of firing temperature on the properties of liquid glass binder (mixture 39).

Temperature, °C	Density, kg/m^3^	Compressive Strength, MPa	Softening Coefficient	Macrostructure
150	830	10.6	0.23	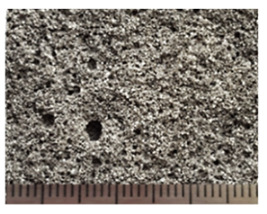
200	710	9.5	0.67	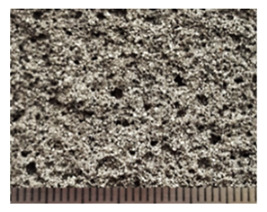
250	590	7.2	0.82	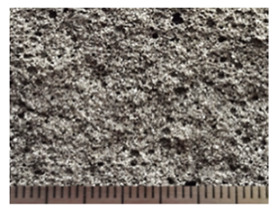
300	585	6.8	0.85	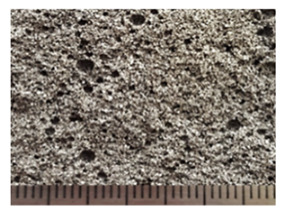
350	570	7.0	0.87	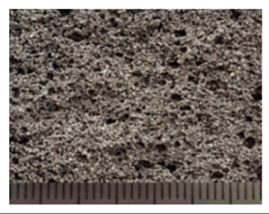

**Table 9 polymers-14-03461-t009:** Main characteristics of the composite material (C_f_ = 0.43).

Characteristics	Value	Appearance
Density, kg/m^3^	500–520	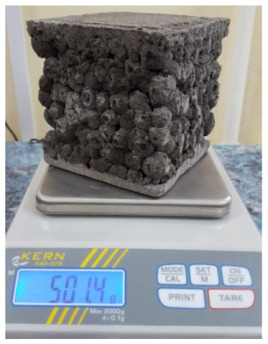
Thermal conductivity, W/(m °C)	0.095–0.100	
Water absorption (48 h), %	17–23	
Compressive strength, MPa	4.7–5.2	
Softening coefficient (7 d)	0.87–0.90	
Frost resistance, cycles	50	
Thermal resistance, thermal exchangers	25	
Heat resistance at a temperature of 1050 °C (“residual” strength after 25 cycles, MPa/initial strength, MPa)	5.2/4.7	

## Data Availability

Not applicable.

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
