# Peer review of "Porous Fly Ash/Aluminosilicate Microspheres-Based Composites Containing Lightweight Granules Using Liquid Glass as Binder"

_polymers, 2022, doi:10.3390/polym14173461_

Round 1

Reviewer 1 Report

1. Authors should mention the materials suppliers and the standards they worked under.

2. Did the authors measure the moisture content of the fly ash before using it? and what was the drying method they used for this purpose, and at what temperature?

3. How was the particles size of the fly ash measured? It is good to add the particles size plot.

4. What is the grinding method used to obtain the spherical shape of the granules of fly ash, or was it random in shape?

5. I prefer to add another paragraph entitled (The tests) to the main paragraph (Materials and methods).

6. Did the authors try to measure the oil absorption as well besides water absorption?

7. Figure 9 has an insufficient resolution. Therefore, replacing it with another one with more resolution is better.

Author Response

Dear Reviewer 1!

Thank you for your interest in my manuscript. Your valuable comments helped make our manuscript even better. All corrections in the manuscript are highlighted in blue.

Comment 1: Authors should mention the materials suppliers and the standards they worked under.

Response: In subsection 2.1, information on suppliers of raw materials has been added.

Item 2.3.5 "The tests" added, which provides information about the standards by which materials are tested

Comment 2: Did the authors measure the moisture content of the fly ash before using it? and what was the drying method they used for this purpose, and at what temperature?

Response: Subsection 2.1 has been supplemented with information on the moisture content of thermal energy waste.

Comment 3: How was the particles size of the fly ash measured? It is good to add the particles size plot.

Response: Figure 2 has been added, which provides information on the content of the main fractions in the composition of thermal energy waste.

Comment 4: What is the grinding method used to obtain the spherical shape of the granules of fly ash, or was it random in shape?

Response: The shape of fly ash particles and aluminosilicate microspheres is formed during high-temperature flare combustion of fuel at coal-fired power plants. The experiments do not provide for the grinding of thermal energy waste.

Comment 5: 5. I prefer to add another paragraph entitled (The tests) to the main paragraph (Materials and methods).

Response:  Added paragraph 2.3.5 "The tests".

Comment 6: Did the authors try to measure the oil absorption as well besides water absorption?

Response: At this stage of research, the authors did not carry out oil absorption by granules and composite materials.

Comment 7: Figure 9 has an insufficient resolution. Therefore, replacing it with another one with more resolution is better.

Response: The resolution of Figure 9 has been increased (10 - according to the new numbering).

Reviewer 2 Report

In this work, authors aimed to create a heat-insulating composite material based on porous granules and a matrix from mixtures of liquid glass and thermal energy waste. The novelty of the work lies in establishing the patterns of formation of a stable structure of a porous material during thermal curing of liquid glass with technogenic fillers. Studies of liquid glass mixtures with different contents of fly ash and aluminosilicate microspheres revealed the possibility of controlling the properties of molding masses in a wide range. To obtain a granular material, liquid-glass mixtures of plastic consistency with a predominance of aluminosilicate microspheres are proposed. The matrix of composite materials is formed by a mobile mixture of liquid glass and a combined filler, in which fly ash predominates. A heat-insulating concrete based on porous aggregate with density of 380-650 kg/m3, thermal conductivity of 0.095-0.100 W/(m ⁰С) and strength of 3.5-9.0 MPa, resistance under conditions of variable values of humidity and temperature have been developed. Overall, this work can be accepted after the following raised concerns are well addressed.

1. Introduction seems very scattering; authors are suggested to adjust the main text structure.

2. In table 5. The scale bar of each granule image should be marked.

3. Are any scientific factors can be used/characterized to replace the subjective description of the liquid glass mixture?

4. The format of references should be consistent, for instance, Ref. 14, 2016; Vol. 156., no page number; Ref. 13/43/45: No DOI number….

Author Response

Dear Reviewer 2!

Thank you for your interest in my manuscript. Your valuable comments helped make our manuscript even better. All corrections in the manuscript are highlighted in blue.

Comment 1: Introduction seems very scattering; authors are suggested to adjust the main text structure.

Response: The "Introduction" section has been supplemented with information about liquid glass materials.

Comment 2: In table 5. The scale bar of each granule image should be marked.

Response: The scale bar of each granule image has been marked in table 5.

Comment 3: Are any scientific factors can be used/characterized to replace the subjective description of the liquid glass mixture?

Response: Subsection 2.1 has been supplemented

Comment 4: The format of references should be consistent, for instance, Ref. 14, 2016; Vol. 156., no page number; Ref. 13/43/45: No DOI number…..

Response: The format of references has been corrected

Round 2

Reviewer 1 Report

My suggestions are included in the article and the rest of the questions are answered convincingly. From my point of view the article can be accepted now.

All the best to the authors

Author Response

Thank you for appreciating our article.